# Respiratory symptoms and cardiovascular causes of deaths: A population-based study with 45 years of follow-up

Knut Stavem[1,2,3]*, Henrik Schirmer[2,4], Amund Gulsvik[5]

1 Department of Pulmonary Medicine, Akershus University Hospital, Lørenskog, Norway, 2 Institute of Clinical Medicine, University of Oslo, Oslo, Norway, 3 Health Services Research Unit, Akershus University Hospital, Lørenskog, Norway, 4 Department of Cardiology, Akershus University Hospital, Lørenskog, Norway, 5 Department of Clinical Science, Faculty of Medicine, University of Bergen, Bergen, Norway

* knut.stavem@medisin.uio.no

**Data Availability Statement:** Data availability: The ethical approval granted by the Regional Committees for Medical and Health Research Ethics in Norway does not allow public sharing of the data. A data set can be made available for

## Abstract

This study determined the association between respiratory symptoms and death from cardiovascular (CV) diseases during 45 years in a pooled sample of four cohorts of random samples of the Norwegian population with 95,704 participants. Respiratory symptoms were assessed using a modification of the MRC questionnaire on chronic bronchitis. We analyzed the association between respiratory symptoms and specific cardiovascular deaths by using Cox regression analysis with age as the time variable, accounting for cluster-specific random effects using shared frailty for study cohort. Hazard ratios (HR) for death were adjusted for sex, highest attained education, smoking habits, occupational air pollution, and birth cohort. Overall, 12,491 (13%) of participants died from CV diseases: 4,123 (33%) acute myocardial infarction, 2,326 (18%) other ischemic heart disease, 2,246 (18%) other heart diseases, 2,553 (20%) cerebrovascular diseases, and 1,120 (9%) other vascular diseases. The adjusted HR (95% confidence interval) for CV deaths was 1.9 (1.7–2.1) in men and 1.5 (1.2–1.9) in women for "yes" to the question "Are you breathless when you walk on level ground at an ordinary pace?". The same item response showed an adjusted HR for death from acute myocardial infarction of 1.8 (1.5–2.1), other ischemic heart disease 2.2 (1.8–2.7), other heart diseases 1.5 (1.1–1.9), cerebrovascular disease 1.8 (1.5–2.3), and other circulatory diseases 1.7 (1.2–2.4). The adjusted HR for CV death was 1.3 (1.2–1.4) when answering positive to the question" Are you more breathless than people of your own age when walking uphill?". However, positive answers to questions on cough, phlegm, wheezing and attacks of breathlessness were after adjustments not associated with early CV deaths. The associations between CV deaths and breathlessness were also present in never smokers. Self-reported breathlessness was associated with CV deaths and could be an early marker of CV deaths.

scientific analysis on request, provided that the respective research institution proofs handling of the data strictly in accordance with ethical regulations (written ethics protocol, full compliance with the Declaration of Helsinki). To ensure full anonymity only the main variables of the final analyses are provided. We confirm that the data file provided constitutes the minimal data set necessary to replicate the findings of our study in their entirety. Data requests can be made to corresponding author Knut Stavem (knut. stavem@medisin.uio.no or data custodian, Haldor Husby (haldor.husby@ahus.no).

**Funding:** The authors received no specific funding for this work.

**Competing interests:** The authors have declared that no competing interests exist.

# Introduction

Respiratory symptoms can be markers of diseases of the heart, blood vessels, lungs, airways and other organs. Indeed, multimorbidity of chronic pulmonary and chronic cardiovascular (CV) diseases is increasingly recognized as a problem worldwide [1].

The symptoms are associated with increased levels of systemic inflammatory markers. Respiratory symptoms have been used as indicators of effects of drugs and rehabilitation [2,3], as well as palliative care in chronic obstructive pulmonary disease (COPD) [4]. Few population studies have focused on the predictive value of respiratory symptoms in deaths of fatal CV deaths using the European (EU) classification of diseases [5]. Previous studies lack statistical power to detect significant risks for subgroups of CV deaths with relatively few events. There is little information available on the association between respiratory symptoms and other heart disease (EU 48), or other circulatory disease (subgroup of EU 50).

During the first 25 years after the introduction of the British Medical Research Council (MRC) questionnaire on chronic bronchitis [6,7], researchers mostly focused on the association of persistent cough and phlegm with death [8–11]. However, after the community study of Tecumseh, Michigan reported an association between dyspnoe and coronary disease [12], more emphasis was put on the association between breathlessness and CV deaths in surveys. An association of breathlessness with all-cause deaths were confirmed in Sweden [13], Denmark [14], United Kingdom [15,16], and for cough and phlegm in Finland [17], but not in Denmark [18].

The first surveys were often small samples of occupational groups or communities of men, above 40 years of age, and the analyses were not adjusted for occupational exposure or educational levels. The questions on respiratory symptoms differed in wording, and the outcome variables were most often all-cause mortality, coronary deaths, or stroke fatalities. Two studies have reported on two subgroups of CV disease, namely death from ischaemic heart disease and stroke [19,20]. Due to the limited population sizes or observation times in previous studies, further research from larger studies is needed to study less common events, as well as events in strata of the populations, such as never smokers.

In two previous studies, we found a consistent, positive association between respiratory symptoms reported and all cause-mortality [21], as well as for COPD and lung cancer as underlying cause of death [22]. As circulatory diseases account for 31% for all-cause mortality in 2012 in Norway [23], we decided to extend the analyses to investigate the long-term relation between respiratory symptoms and mortality due to all EU subgroups of CV diseases except for lung embolism. The risk of respiratory symptoms on subgroups on CV mortality was adjusted for gender, age groups, smoking habits, occupational exposure to air pollution as well as highest -attainable education and self-reported heart and lung diseases.

# Materials and methods

## Study population

This multicohort study used harmonized individual-level data from four cross-sectional surveys in the City of Oslo 1972 and 1998–1999, Hordaland County in 1985, 1988–1990 (including Sauda municipality in Rogaland County) and 1998–1999 [24]. Oslo, the capital of Norway, had 477,476 inhabitants in 1972 and 499,693 persons in 1998. Hordaland County is a combined rural and urban (Bergen) population with 399,702 persons in 1985, 405,063 in 1988 and 428,823 in 1998.

The target population was born 1895–1982. The sample frames were updated lists from the Norwegian National Population Registry. Invitees were drawn at random for the 1972 (Oslo

**Table 1. Flowchart of randomly sampled individuals in the study of respiratory symptoms and cardiovascular deaths in Norway.**

| | Oslo cohort 1972 | Hordaland county cohort 1985 | Hordaland county and Sauda municipality Rogaland county 1988–1990 | Hordaland and Oslo counties cohort 1998–1999 | Total |
|---|---|---|---|---|---|
| Sample drawn from target populations | 19998 | 4992 | 112235 | 25000 | 162225 |
| Present in previous samples, excluded | 1* | 2 | 1893 | 1627 | 3523 |
| Eligible, unique persons | 19997 | 4990 | 110342 | 23373 | 158702 |
| Eligible after missing times excluded | 19892 | 4982 | 108812 | 23210 | 156896 |
| Respondents (response to at least 1 of 17 questionnaire items) | 17690 | 4461 | 77003 | 15870 | 115024 |
| Respondents to smoking habits | 17680 | 4404 | 76675 | 15623 | 114380 |
| Respondents to smoking habits and education | 17377 | 4347 | 75406 | 14994 | 112124 |
| Respondents to smoking habits, education and occupational exposure | 16445 | 4307 | 71958 | 14765 | 107475 |
| Respondents to all items on respiratory symptoms | 16084 | 4137 | 69168 | 14492 | 103881 |
| Responded to three items on previous heart disease | 15421 | 3785 | 63540 | 12958 | 95704 |

*removed duplicate record.

72), 1985 (Hordaland 85) and 1998–1999 (Oslo/Hordaland 98–99) surveys. The random sampling was conducted by the national Population register using a random number generator. The 1988–1990 survey (Støvlunge 88–90) invited all men born 1914–1959, plus a 10% sample of the general population of City of Bergen born 1895–1950 and examined in 1965–1970 [25], leading to about 6% women in this cohort. We excluded 3,523 persons already included in one cohort from later cohorts, and the eligible population sample comprised 158,702 unique persons (Table 1). We have previously presented the recruitment and pooling of these cohorts [21].

In all surveys, we used a self-administered questionnaire filled out at home by the person invited, without any help from persons in the survey center. In the analyses, we included respondents who provided information on smoking, education, occupational exposure to air pollution, respiratory symptoms, as well as additional information previous heart diseases, in total 95 704 persons; 60% of the sample of unique persons (Table 1).

The start of the observation was the date of returning the fill out self-administered questionnaire to the survey center. However, the actual date was not available in two of the cohorts. In one cohort we had the month and year of participation, and in the fourth we used a fixed date approximately when invitations were sent.

## Ethical approval and consent

Regional committees for Medical Research Ethics were established in Norway in 1985 [26]. Therefore, for the first survey cohort in 1972 (Oslo 72) there was no requirement for ethical approval. Answering and returning the questionnaire was considered as an informed consent.

The participants of the Hordaland County cohort 1985 survey (Hordaland 85) and the Hordaland and Oslo counties cohort 1998–1999 survey (Oslo/Hordaland 98–99) all signed an informed consent, which was approved by the Regional Ethical Committee of Health Region West [24].

The attendants of the Hordaland County and Sauda municipality in Rogaland County 1988–1990 survey (Støvlunge 88–90) were invited to X-ray surveillance for tuberculosis where

attendance is demanded by Norwegian law. The consent information described the health examination, including risk of technical procedures and blood collections, as well as data protection and genetic assessment. Participants could ask questions about the examination, and they were required to give written consent before any part of the health examination was conducted [24].

The Norwegian Data Inspectorate allowed the use of unique personal identification numbers to gather all information collected in the four surveys on each individual and to link information from Statistics of Norway on death, causes of deaths and highest attained education. The Norwegian Directorate of Health and Social Services granted us permission to receive information on notification of causes of deaths. Finally, the Data Inspectorate, the Norwegian Directorate of Health and Social Services and Regional Ethical Committee granted permission to have a common data registry for all surveys, for all follow-up studies, and the specific causes of deaths [24].

The study was approved by the Regional Committee on Medical Research Ethics (reference 2017/1679), the Norwegian Data Inspectorate (07/00414), and the Norwegian Directorate of Health (07/949).

## Questionnaire

We used a modification of a questionnaire on respiratory symptoms, which was approved by MRC Committee on research into Chronic Bronchitis in 1966 [7].

The questionnaire included 11 questions (Table 2), covering current cough, phlegm, wheezing, periods of cough and/or breathlessness with and without exercise. This questionnaire has previously been presented [20–22,24,27]. The questions were aggregated into three symptom groups: breathlessness on exercise scored 0 to 4; cough and/or phlegm symptoms scored 0 to 5, and attacks of breathlessness and wheezing scored 0 to 2 [21,22]. A higher score represents more severe symptoms. The validity of the Norwegian respiratory questionnaire has been evaluated [28] and compared with the original MRC questionnaire [27].

**Table 2. Questions (Q) on respiratory symptoms and scores.**

| Respiratory symptom group | Questions, score |
|---|---|
| Cough and phlegm (Bronchitis like symptoms), scores 0–5 | Q 8. Do you usually cough and clear your throat in the morning? Yes = 1, no = 0. |
| | Q 9. Do you usually cough during the day? Yes = 1, no = 0. |
| | Q 10. When you cough or clear your throat do you usually bring up phlegm? Yes = 1, no = 0. |
| | Q 11. Do you have cough for 3 months or more altogether during a year? Yes = 1, no = 0. |
| | Q 12. During the last 2 years, have had a cough and/or phlegm in connection with a cold for more than 3 weeks? Yes = 1, no = 0. |
| Attacks of breathlessness and wheeze (Asthma like symptoms), scores 0–2 | Q 17. Do you have attacks of breathlessness? Yes = 1, no = 0. |
| | Q 18.Have you ever had wheezing in your chest? Yes = 1, no = 0. |
| Breathless, scores 0–4 | Q 13. Are you more breathless than people of your own age when walking uphill? Yes = 1, no = 0. |
| | Q 14. Are you breathless when you climb two flights of stairs at an ordinary pace? Yes = 1, no = 0. |
| | Q 15. Are you breathless when you walk on level ground at an ordinary pace? Yes = 1, no = 0 |
| | Q 16. Are you breathless when at rest? Yes = 1, no = 0 |

The questionnaire also included questions on smoking history and occupational exposure to air pollution [29]. Smoking was categorized as current smoker (daily at the time of the study), ex-smokers or never smokers. Occupational exposure to air pollution was defined by responding "yes" to "Have you been exposed to particulates, gases or damp at your working place?"

Self-report of cardiac diseases was defined as a positive reply to the question: Have you at any time been treated by a physician or have you ever been in hospital because of a) heart infarction, b) angina pectoris, and/or c) other heart diseases? Self-report of pulmonary diseases was defined as positive reply to the question: Have you at any time been treated by a physician or have you ever been in hospital because of a) bronchitis, b) asthma, c) emphysema and/or tuberculosis? A negative or a do not know answer was defined as a negative answer.

## Follow-up and census data

Date of death and emigration until December 31, 2016 were obtained from the NorwegianCause of Death Registry. Educational attainment was obtained from the national census for each decade and grouped according to the maximum length of education using three levels: compulsory education (7–10 years), medium level (11–13 years) and university level (≥14 years).

All inhabitants of Norway have a unique personal identification number that allows complete follow-up until death or emigration. In total 156,896 people were people were observed in the pooled cohorts; median follow-up was 26.7 years (range 1 day to 45.2 years). The cumulative time for observation in the total sample was 3,541,219 person-years. A total of 95,704 respondents with known smoking status that had responded to items on respiratory symptoms and previous heart disease represented 2,285,97 person-years at risk, of whom 32,583 were never smokers and represented 813,565 person-years at risk.

## Classification of causes of death

The Norwegian Cause of Death Registry has directly identifiable data on individuals from 1951 [30]. It covers about 98% of all deaths in Norway, and most missing values are for people dying abroad. Until 2005 the underlying cause of death was determined manually by the coding personnel at the Cause of Death Registry. From 2005 the underlying cause of death has been determined electronically by a software application, Automated Classification of Medical Entities (ACME), developed by the National Center for Health Statistics in the US. The listed causes of death on the death certificate are first translated manually into ICD-10 codes, and then the ACME software determines the underlying cause of death [31–33].

The registry classifies deaths according to the European Shortlist for causes of death (EU) [5,34]. The classification of CV causes as underlying cause (Table 3) includes acute myocardial infarction (EU 46) and other ischaemic heart disease (EU 47) i.e. angina pectoris and heart

Table 3. Classification of underlying cardiovascular causes of death.

| Cause | EU 2012 classification | ICD-8 | ICD-9 | ICD-10 |
|---|---|---|---|---|
| Acute myocardial infarction | 46 | 410–411 | 410–411 | I21.0-I21.9, I22.0-I22.9 |
| Other ischemic heart disease | 47 | 412–414 | 412–414 | I20,I24-I25 |
| Other heart diseases | 48 | 421–429 | 421–429 | I31-I51 |
| Cerebrovascular | 49 | 430–438 | 430–438 | I60-I69 |
| Other circulatory | 50 (excl. lung embolism) | 390–405, 420–429, 440–448, 451–458 (excl. 415.1) | 390–405, 415–416, 420–429, 440–449, 451–459 (excl. 415.1) | I05-I13,I26-I28, I70-I99 (excl. I26.9) |

aneurysm which can be combined to ischaemic heart disease (EU 46–47), other heart diseases (E48), cerebrovascular disease (E 49) and other diseases of the circulatory system (EU 50) [35]. The category "Other diseases of the circulatory system" was divided into lung embolism (EU 50a) and others (EU 50b). Ischemic heart disease (EU 46–47) are diseases causing less blood and oxygen to the heart musculature. Other heart diseases (EU 48) comprises heart failure, non-rheumatic valvular diseases, heart arrhythmia, inflammations of the heart (pericarditis, endocarditis, myocarditis) and cardiomyopathy. Cerebrovascular deaths (EU 49) may be caused by cerebral haemorrhage, cerebral embolism, arteriosclerosis of the cerebral arteries, or aneurysm [36]. Other circulatory diseases (EU 50b exclusive lung embolism) includes diseases such as chronic rheumatic valvular diseases, hypertensive heart disease, pulmonary heart/vessel diseases, arteriosclerosis, aortic aneurysm, and other diseases of the arterial, venous and lymphatic system.

Death certificates are the only source of information on death in >90% of cases. Autopsy rates in Norway have fluctuated over time.

In the early 1960 it was about 10% of all deaths, increasing to 18% in 1986 and gradually declining to slightly above 7% in 2010 [32]. The rate was between 7 and 9% from 2011 to 2019 [33]. The diagnostic validity of cerebral stroke and ischemic heart disease as underlying cause of death in this registry showed a substantial agreement between mortality statistics and autopsy findings, although there was a selection of cases for post-mortem examinations [37].

## Statistical analyses

Descriptive statistics are presented as number (%). For start dates for observations in the cohorts, we used 5 October 1972 in the Oslo county 1972 cohort, the 15th of the actual starting month in the Hordaland county and Sauda municipality cohort 1988–1990, and the actual start date in the other two cohorts. Other missing start dates for participants were imputed using the median start date in the same cohort: 31 December 1989 in the Hordaland county and Sauda municipality cohort 1988–1990 (n = 100) and 5 October 1998 in the Oslo and Hordaland counties cohort 1998–1999 (n = 4).

The cohort members were followed until death or censored at the date of emigration or end of follow-up on 31 December 2016, whichever came first. For some people that emigrated (n = 231), we did not have a date of emigration, but only an interval. These cases were censored at the mid-point of the interval [21]. We did not impute missing values for other variables.

We pooled the four cohorts and analyzed the association of symptom scores of breathlessness on effort, cough or/and phlegm and attacks of breathlessness or/and wheeze with cause-specific mortality. We used Cox proportional hazards analysis, with age as the time variable [38]. The analyses were prepared using shared frailty for study cohort, i.e. incorporating cluster-specific random effects to account for within-cluster homogeneity in outcomes [39].

All analyses were multivariable, adjusting for sex, education (<10, 11–13, ≥14 years), smoking habits (never, previous, current smoker), occupational exposure (dust/fume vs. none), birth cohort (1895–1919, 1920–1929, 1930–1939, 1940–1949, 1950–1959, 1960–) and self-reported history of CV disease (angina pectoris, myocardial infarction, or other heart disease). The results are presented as hazard ratios (HR) of death with 95% confidence intervals (CI).

In addition, we conducted analyses in strata of the pooled sample (women, men, never smokers) using a similar approach with the same covariates.

Finally, we repeated the same analysis in the pooled sample after excluding (1) those with self-reported heart disease (angina pectoris, myocardial infarction or other heart disease) with the same covariates as in the primary analysis, and (2) those with self-reported heart disease or lung disease (bronchitis, asthma, emphysema and/or tuberculosis).

The proportional hazards assumption was checked graphically using log-log plots and was considered acceptable. We used Stata version 17.0 (StataCorp, College Station, TX, USA) for all analyses. We chose a 5% significance level with two-sided tests.

## Results

The target population comprised 156,896 persons, while altogether 95,704 (61%) responded to the relevant items and were included in the analysis (S1 Table). The sample of persons analysed compared with persons in the four target populations differed by being slightly older, fewer with only compulsory education but without sex differences. However identical smoking habits and occupational exposure were observed in those who answered or not answered a self-administered questionnaire on respiratory symptoms. The percentages of participants in the analysed sample compared with the target population was slightly higher for the two first surveys (Oslo 72, Hordaland 85) compared with the two last surveys (Støvlunge 88–90, Oslo/Hordaland 98–99) The overall representativeness of the analysed sample for the target population was excellent.

Respiratory symptoms were prevalent in the adult Norwegian population, as 19% in our sample of 95,704 participants without self-reported heart disease reported one or more symptoms of breathlessness, 34% had one or more symptoms on cough and phlegm and 23% had attacks of breathlessness and/or wheeze (S2 Table).

Per 31 December 2016, 12,490 persons (13%) had died from CV diseases, 10,229 men and 2,261 women. The causes of CV deaths were: 4,123 (33%) acute myocardial infarction (EU 46), 2,326 (18%) with other ischemic heart diseases (EU 47), 2,246 (18%) other heart diseases (EU 48), 2,553 (20%) with cerebrovascular diseases (EU 49), 124 (1%) with lung embolism (ICD-10 I26.9) and 1.120 (9%) with other circulatory diseases (EU 50 exclusive lung embolism).

### Acute myocardial infarction and other ischemic heart diseases

The hazard ratio (HR) for death showed a similar pattern for deaths of acute myocardial infarction and deaths of other ischemic diseases (Table 4).

An increasing risk of deaths due to ischaemic heart disease was observed with higher breathlessness score until breathless score 3. An affirmative answer to Q16: Are you breathless when at rest? (score 4), predicted a lower risk of ischaemic heart disease death than the affirmative question Q 15 Are you breathless when you walk on level ground at an ordinary pace. Increasing score of cough and phlegm score did not show any overt association with deaths due to ischaemic heart disease. Surprisingly death due acute myocardial infarction declined with increasing score of attacks of breathlessness and wheeze, but this was not observed for deaths due to other ischaemic diseases

### Other heart diseases

A trend of higher mortality due to other heart diseases (Table 4) was observed with increasing score of breathlessness on effort, while breathlessness at rest had a lower risk than when walking on level ground. However, a cough phlegm score with yes to 3 or more questions was borderline associated with deaths due to cardio vascular diseases. The trends of risks for deaths of other heart diseases did not differ from the deaths of ischemic heart diseases.

### Cerebrovascular disease

The risk of stroke deaths increased with breathlessness score (Table 4). The risk of cough and phlegm for lethal stroke was not observed in any dose-response way. However, unexpectedly

**Table 4. Hazard ratios with 95% confidence intervals and p-values according to cause of death, multivariable proportional hazards regression analysis with adjustment for sex, education, smoking status, occupational exposure to gas/dust, previous angina pectoris, previous other heart disease and previous myocardial infarction, and birth cohort (n = 95704).**

| | All cardiovascular | | Acute MI | | Other ischemic heart | | Other heart | | Cerebrovascular | | Other circulatory | |
|---|---|---|---|---|---|---|---|---|---|---|---|---|
| | HR | 95%CI | HR | 95%CI | HR | 95%CI | HR | 95%CI | HR | 95%CI | HR | 95%CI |
| Breathless on effort, score (vs. 0) | | | | | | | | | | | | |
| 1 | 1.30*** | [1.23,1.38] | 1.30*** | [1.19,1.43] | 1.50*** | [1.32,1.69] | 1.17* | [1.02,1.34] | 1.31*** | [1.15,1.48] | 1.23* | [1.02,1.48] |
| 2 | 1.50*** | [1.41,1.60] | 1.47*** | [1.32,1.64] | 1.73*** | [1.51,2.00] | 1.51*** | [1.29,1.76] | 1.32*** | [1.13,1.55] | 1.51*** | [1.21,1.88] |
| 3 | 1.81*** | [1.64,1.99] | 1.75*** | [1.50,2.05] | 2.19*** | [1.80,2.66] | 1.46** | [1.12,1.91] | 1.83*** | [1.46,2.30] | 1.68** | [1.19,2.36] |
| 4 | 1.64*** | [1.38,1.95] | 1.56** | [1.16,2.09] | 1.75** | [1.20,2.54] | 1.10 | [0.66,1.84] | 2.12*** | [1.46,3.09] | 2.11** | [1.26,3.54] |
| Cough and phlegm, score (vs. 0) | | | | | | | | | | | | |
| 1 | 0.98 | [0.93,1.03] | 0.99 | [0.90,1.07] | 1.04 | [0.93,1.17] | 0.97 | [0.86,1.10] | 0.94 | [0.84,1.05] | 0.87 | [0.73,1.03] |
| 2 | 1.04 | [0.97,1.12] | 1.08 | [0.96,1.22] | 1.23** | [1.05,1.42] | 0.97 | [0.82,1.16] | 0.91 | [0.77,1.07] | 0.92 | [0.73,1.16] |
| 3 | 1.10* | [1.01,1.20] | 0.95 | [0.81,1.11] | 1.07 | [0.87,1.31] | 1.24* | [1.01,1.54] | 1.26* | [1.05,1.53] | 1.12 | [0.85,1.47] |
| 4 | 1.01 | [0.91,1.13] | 0.84 | [0.70,1.02] | 1.02 | [0.80,1.29] | 1.26 | [0.98,1.61] | 1.05 | [0.82,1.35] | 1.22 | [0.89,1.66] |
| 5 | 1.10 | [0.98,1.24] | 0.87 | [0.87,1.31] | 1.05 | [0.80,1.38] | 1.41* | [1.06,1.88] | 1.00 | [0.75,1.34] | 0.93 | [0.62,1.40] |
| Attacks of breathlessness and wheeze, score (vs. 0) | | | | | | | | | | | | |
| 1 | 0.97 | [0.92,1.02] | 1.02 | [0.94,1.11] | 0.95 | [0.85,1.06] | 0.99 | [0.88,1.12] | 0.85** | [0.75,0.95] | 1.10 | [0.93,1.29] |
| 2 | 0.93 | [0.86,1.01] | 0.92 | [0.80,1.05] | 1.06 | [0.89,1.25] | 0.87 | [0.71,1.07] | 0.83 | [0.69,1.01] | 1.01 | [0.77,1.31] |

* p<0.05

** p<0.01

*** p<0.001.

an inverse association was observed between stroke mortality and a single asthmatic symptom but not for combination of symptoms.

## Other circulatory diseases

The risk factors of respiratory symptoms for lethal other circulatory diseases (excl. lung embolism) were similar to deaths for all CV diseases (Table 4).

## Stratified analyses

**Women.** Altogether 20,221 women were included in this analyse (Table 5). A clear association and an increased risk of CV deaths was observed with breathlessness at lower level of exercise. The multivariable proportional hazard regression showed also slightly less risk for CV death in the presence of self-report of both wheezing and attacks of breathlessness. The load of cough and phlegm symptoms did not show any significant risk for CV deaths. The patterns of risk factors for early CV deaths in women was similar for the five specific CV causes of deaths.

**Men.** Altogether 75,483 men were included in this analysis, and the pattern of association between respiratory symptoms and CV deaths in men was similar to that in women (Table 5).

The risk of deaths due to CV for each exercise level of breathlessness score was higher in men than women. Cough and phlegm were not associated with increased risk of CV deaths.

No consistent association was observed between cough and phlegm scores or attacks of breathlessness and wheezing with deaths of all CV diseases or subgroups of CV diseases.

**Never smokers.** In the group of 32,583 never smokers (Table 6) there was a trend of increasing HR with breathless score from 0 to 3, while no significant association was observed

**Table 5. Stratified analyses for women and men.** Hazard ratios with 95% confidence intervals and p-values according to cause of death, multivariable proportional hazards regression analysis with adjustment for education, smoking status, occupational exposure to gas/dust, previous angina pectoris, previous other heart disease and previous myocardial infarction, and birth cohort.

| | All cardiovascular | | Acute myocardial infarction | | Other ischemic heart | | Other heart | | Cerebrovascular | | Other circulatory | |
|---|---|---|---|---|---|---|---|---|---|---|---|---|
| | HR | 95%CI | HR | 95%CI | HR | 95%CI | HR | 95%CI | HR | 95%CI | HR | 95%CI |
| *Women (n = 20,221)* | | | | | | | | | | | | |
| Breathless on effort, score (vs. 0) | | | | | | | | | | | | |
| 1 | 1.15* | [1.00,1.32] | 0.96 | [0.73,1.27] | 1.37 | [0.97,1.94] | 0.98 | [0.71,1.34] | 1.30* | [1.02,1.67] | 1.35 | [0.88,2.08] |
| 2 | 1.36*** | [1.17,1.58] | 1.31 | [0.99,1.74] | 1.90*** | [1.33,2.72] | 1.33 | [0.95,1.86] | 1.14 | [0.84,1.55] | 1.24 | [0.73,2.10] |
| 3 | 1.51*** | [1.20,1.91] | 1.81** | [1.23,2.64] | 1.03 | [0.52,2.01] | 1.70 | [0.99,2.93] | 1.45 | [0.91,2.32] | 1.08 | [0.45,2.60] |
| 4 | 1.32 | [0.91,1.92] | 1.01 | [0.50,2.03] | 1.07 | [0.40,2.92] | 1.98 | [0.84,4.71] | 1.62 | [0.78,3.34] | 1.59 | [0.51,4.90] |
| Cough and phlegm, score (vs. 0) | | | | | | | | | | | | |
| 1 | 0.92 | [0.82,1.04] | 0.95 | [0.76,1.20] | 1.00 | [0.73,1.36] | 0.90 | [0.68,1.18] | 0.93 | [0.74,1.17] | 0.75 | [0.49,1.15] |
| 2 | 1.15 | [0.98,1.35] | 1.26 | [0.94,1.70] | 1.34 | [0.89,2.00] | 1.22 | [0.85,1.74] | 0.96 | [0.68,1.34] | 0.97 | [0.57,1.66] |
| 3 | 1.15 | [0.94,1.40] | 1.06 | [0.71,1.57] | 1.07 | [0.62,1.86] | 1.23 | [0.79,1.91] | 1.38 | [0.97,1.97] | 0.99 | [0.51,1.89] |
| 4 | 1.27 | [0.98,1.65] | 1.24 | [0.77,2.01] | 1.46 | [0.76,2.79] | 1.38 | [0.75,2.54] | 1.35 | [0.81,2.25] | 0.88 | [0.34,2.24] |
| 5 | 1.05 | [0.77,1.41] | 1.07 | [0.61,1.85] | 0.73 | [0.30,1.78] | 0.97 | [0.46,2.02] | 1.47 | [0.86,2.50] | 0.6 | [0.20,1.79] |
| Attacks of breathlessness and wheeze, score (vs. 0) | | | | | | | | | | | | |
| 1 | 0.92 | [0.81,1.05] | 0.84 | [0.66,1.07] | 0.98 | [0.71,1.36] | 0.88 | [0.66,1.18] | 0.83 | [0.64,1.06] | 1.54* | [1.05,2.27] |
| 2 | 0.80* | [0.65,0.99] | 0.97 | [0.67,1.39] | 0.81 | [0.47,1.39] | 0.67 | [0.40,1.10] | 0.64* | [0.41,0.98] | 1.26 | [0.66,2.42] |
| *Men (n = 75,483).* | | | | | | | | | | | | |
| Breathless on effort, score (vs. 0) | | | | | | | | | | | | |
| 1 | 1.34*** | [1.26,1.42] | 1.37*** | [1.24,1.52] | 1.52*** | [1.34,1.74] | 1.22** | [1.05,1.43] | 1.30*** | [1.13,1.51] | 1.2 | [0.97,1.48] |
| 2 | 1.53*** | [1.42,1.64] | 1.49*** | [1.32,1.68] | 1.70*** | [1.46,1.98] | 1.54*** | [1.29,1.84] | 1.39*** | [1.16,1.66] | 1.57*** | [1.24,2.00] |
| 3 | 1.85*** | [1.66,2.06] | 1.69*** | [1.42,2.01] | 2.34*** | [1.91,2.88] | 1.42* | [1.04,1.92] | 1.96*** | [1.51,2.55] | 1.81** | [1.25,2.63] |
| 4 | 1.70*** | [1.40,2.06] | 1.62** | [1.17,2.25] | 1.84** | [1.23,2.76] | 0.91 | [0.48,1.73] | 2.34*** | [1.50,3.63] | 2.15* | [1.20,3.88] |
| Cough and phlegm, score (vs. 0) | | | | | | | | | | | | |
| 1 | 0.99 | [0.94,1.05] | 0.99 | [0.90,1.09] | 1.06 | [0.94,1.19] | 0.99 | [0.87,1.13] | 0.94 | [0.83,1.07] | 0.9 | [0.75,1.08] |
| 2 | 1.02 | [0.94,1.10] | 1.06 | [0.93,1.20] | 1.21* | [1.03,1.42] | 0.91 | [0.74,1.11] | 0.89 | [0.73,1.07] | 0.9 | [0.70,1.17] |
| 3 | 1.09 | [0.99,1.21] | 0.94 | [0.79,1.12] | 1.08 | [0.87,1.34] | 1.24 | [0.98,1.58] | 1.22 | [0.98,1.53] | 1.16 | [0.85,1.58] |
| 4 | 0.97 | [0.87,1.09] | 0.79* | [0.64,0.98] | 0.99 | [0.76,1.27] | 1.23 | [0.93,1.62] | 0.98 | [0.74,1.29] | 1.3 | [0.94,1.81] |
| 5 | 1.11 | [0.98,1.27] | 1.07 | [0.86,1.34] | 1.1 | [0.83,1.46] | 1.54** | [1.13,2.11] | 0.87 | [0.61,1.24] | 1 | [0.64,1.56] |
| Attacks of breathlessness and wheeze, score (vs. 0) | | | | | | | | | | | | |
| 1 | 0.98 | [0.93,1.03] | 1.05 | [0.96,1.14] | 0.95 | [0.84,1.07] | 1.01 | [0.88,1.15] | 0.85* | [0.74,0.97] | 1.02 | [0.86,1.22] |
| 2 | 0.96 | [0.88,1.04] | 0.91 | [0.78,1.06] | 1.09 | [0.91,1.30] | 0.93 | [0.74,1.16] | 0.9 | [0.73,1.12] | 0.96 | [0.72,1.29] |

* p<0.05

** p<0.01

*** p<0.001.

for breathless score 4. However, cough and phlegm were not a risk factor for death from CV diseases. The associations between self-reported previous clinical diagnoses of heart diseases and CV deaths were similar in never smokers as in the total population.

**Persons without self-reported heart and/or lung diseases.** After excluding persons with known heart disease at baseline the sample consisted of 90,316 persons (S3 Table). The association of breathlessness on effort with the increased risk of CV deaths was unanimous. However, neither cough/phlegm score nor attacks of breathless and wheezing score were associated with CV deaths.

**Table 6. Never-smokers.** Hazard ratios with 95% confidence intervals and p-values according to cause of death, multivariable proportional hazards regression analysis adjusted for sex, education, occupational exposure to gas/dust, previous angina pectoris, previous other heart disease and previous myocardial infarction, and birth cohort (n = 32583).

| | All CV | | Acute MI | | Other ischemic heart | | Other heart | | Cerebrovascular | | Other circulatory | |
|---|---|---|---|---|---|---|---|---|---|---|---|---|
| | HR | 95%CI | HR | 95%CI | HR | 95%CI | HR | 95%CI | HR | 95%CI | HR | 95%CI |
| Breathless on effort, score (vs. 0) | | | | | | | | | | | | |
| 1 | 1.19** | [1.05,1.34] | 1.09 | [0.88,1.36] | 1.65*** | [1.27,2.16] | 1.20 | [0.92,1.56] | 1.07 | [0.83,1.39] | 1.10 | [0.69,1.77] |
| 2 | 1.52*** | [1.33,1.74] | 1.49** | [1.17,1.88] | 2.31*** | [1.73,3.09] | 1.48** | [1.10,1.98] | 1.18 | [0.87,1.59] | 1.18 | [0.67,2.07] |
| 3 | 2.06*** | [1.68,2.54] | 2.23*** | [1.59,3.13] | 3.14*** | [2.04,4.82] | 1.39 | [0.79,2.45] | 1.76* | [1.08,2.87] | 1.78 | [0.74,4.26] |
| 4 | 0.97 | [0.59,1.58] | 0.67 | [0.25,1.83] | 2.15 | [0.92,5.01] | 1.07 | [0.39,2.98] | 0.61 | [0.15,2.51] | 0.85 | [0.11,6.35] |
| Cough and phlegm, score (vs. 0) | | | | | | | | | | | | |
| 1 | 0.89* | [0.81,0.99] | 0.94 | [0.78,1.13] | 0.89 | [0.69,1.15] | 0.86 | [0.69,1.07] | 0.89 | [0.72,1.11] | 0.86 | [0.59,1.26] |
| 2 | 1.09 | [0.92,1.29] | 0.83 | [0.59,1.18] | 1.87*** | [1.35,2.60] | 1.09 | [0.76,1.56] | 0.92 | [0.63,1.35] | 1.01 | [0.53,1.93] |
| 3 | 1.00 | [0.80,1.24] | 0.98 | [0.65,1.47] | 1.16 | [0.69,1.92] | 0.92 | [0.57,1.49] | 1.08 | [0.69,1.69] | 0.93 | [0.41,2.14] |
| 4 | 0.95 | [0.68,1.32] | 0.87 | [0.46,1.65] | 1.18 | [0.57,2.45] | 0.67 | [0.29,1.52] | 1.36 | [0.73,2.51] | 0.75 | [0.18,3.09] |
| 5 | 0.80 | [0.55,1.17] | 0.86 | [0.44,1.69] | 0.49 | [0.15,1.54] | 1.08 | [0.55,2.14] | 0.73 | [0.30,1.78] | 0.42 | [0.06,3.11] |
| Attacks of breathlessness and wheeze, score (vs. 0) | | | | | | | | | | | | |
| 1 | 0.99 | [0.88,1.11] | 1.02 | [0.83,1.26] | 0.95 | [0.73,1.24] | 1.08 | [0.84,1.38] | 0.87 | [0.68,1.12] | 1.13 | [0.74,1.74] |
| 2 | 0.95 | [0.77,1.16] | 0.96 | [0.66,1.40] | 0.87 | [0.55,1.38] | 1.21 | [0.80,1.84] | 0.70 | [0.43,1.14] | 1.15 | [0.53,2.47] |

* p<0.05

** p<0.01

*** p<0.001.

When excluding all persons with known lung diseases (bronchitis, asthma, emphysema and/or tuberculosis) in addition to known heart disease (n = 72,339) the associations between respiratory symptoms and CV deaths were essentially unchanged (S4 Table).

## Discussion

### Main findings

This is the first study showing a consistent association between graded breathlessness and five different causes of deaths within CV diseases after adjusting for sex, age, birth cohort, highest attained education, smoking habits, occupational exposure to air pollution and previous clinical diagnosis of heart diseases. The associations increased with the severity of breathlessness until score 3, and the pattern of associations was similar in women and in never smokers.

Breathlessness showed a higher HR for ischaemic heart disease than for other heart diseases, which include inflammatory heart diseases such as endocarditis, myocarditis, and cardiomyopathies. The cough and phlegm score, as well as the attacks of breathlessness and wheezing score, had none or little value for predicting subgroups of CV deaths. A high attainable level of education was a major and consistent predictor of reduced risk of CV death.

This study comprises a very large pooled cohort with long follow-up, enabling us to present new data on five causes of specific deaths from CV diseases. It would not be possible or meaningful to detect some of the associations in smaller cohorts. This explains why there is little previous information on the association between respiratory symptoms and the less common causes of CV deaths.

Cause of breathlessness in heart diseases may be myocardial ischaemia causing transient episodes of left ventricular end-diastolic as well as left atrial pressure increase with subsequent

transient or long-term pulmonary engorgement. This leads to increase airways resistance, and may come before any changes in pulmonary compliance [40,41]. Accordingly, the revised Diamond and Forrester table of pre-test probabilities of obstructive coronary disease in the European Society of Cardiology guidelines on chronic coronary syndromes 2019 lists increasing risk of obstructive coronary artery disease with dyspnoea only or dyspnoea as the primary symptom [42–44].

That there was less association of the CV outcomes between breathlessness score 4 than score 3, might be related to one of the questions "Are you breathless when at rest?". It is possible that this item captures other types of deconditioning than those related to CV risk, such as anxiety or hyperventilation.

Cough and phlegm are signs of chronic infection of the bronchial mucosa. Although infections may play some role in the development of coronary disease, a causal verification is lacking. A medical history of chronic lung disease predicted 22% lower risk of obstructive coronary artery disease in a study of elective coronary angiography from the American College of Cardiology National Cardiovascular Data Registry [45].

## Comparison with other studies

The predictive value of a simple, postal questionnaire on CV symptoms for detection of early deaths of coronary diseases was observed more than 40 years ago [8,10], but they did not adjust for other respiratory symptoms like breathlessness. However, a community study of Tecumseh, Michigan showed that the mortality rate of coronary heart disease was twice the rate for cough and phlegm in men and women [12].

The present study supports and extends the previous findings in one of the four cohorts with 30 years follow up [20], which showed a strong association between respiratory symptoms and deaths due to ischaemic heart disease and a weaker association between respiratory symptoms and stroke.

The Nord-Trøndelag county study in Norway of 9,462 persons followed for 14 years did not find an association between respiratory symptoms and CV mortality [46]. The respiratory questions were more or less identical to those in the present study, and the analyses were adjusted for age, smoking status, BMI, self-reported physical activity, education, and lung function but not for occupational exposure. Their population at risk was only a tenth of the present study, and there were no analyses of subgroups of CV deaths.

The Busselton study in Australia [19] investigated the association between respiratory symptoms and deaths due to coronary and stroke diseases with 26 years of follow up of about 4,300 subjects. The study dichotomized the respiratory symptoms, whereas the present study had graded responses of several categories of symptoms. However, the Busselton study included lung function and several CV risk factors in the analysis, finding that only dyspnoea was associated with coronary deaths in men and women.

A recent population-based survey in the US, with 10,881 persons followed for an average of 19 years, reported that persons with mild or moderate/severe dyspnoea, but free of previous cardiopulmonary disease, had an increased risk of incident heart failure and myocardial infarction [47].

A small population-based survey in Sweden of birth cohort 1914, with 699 men and 99% follow up of deaths, reported that continuous and incident breathlessness was associated with cardiac events, however this was statistically non-significant after adjusting for competing events and confounders [48].

A population-based survey in 2015–2016 in Tromsø, Norway included 1,539 people >40 years of age. Heart failure was defined by echocardiography [49] and chronic obstructive

pulmonary disease by spirometry [4], and a questionnaire recorded respiratory symptoms. The prevalence of heart failure was 9% and of COPD was 5%, but only 10% of those with heart failure had COPD [50].

Wheezing and attacks of breathlessness can be associated with allergic inflammation. Allergy has not been associated with coronary disease. No associations were observed between these symptoms and deaths due to the five causes of CV diseases in our study.

Breathlessness on effort without cough and phlegm is in a clinical setting characteristic for emphysema and not chronic obstructive bronchitis, but also of obstructive coronary artery disease as recently shown [43]. Although we have also focused on participants without self-reported cardiac disease, we cannot exclude the possibility of dyspnoea being an expression of undiagnosed CV disease. In a survey CT coronary angiography of an asymptomatic general population in Sweden, more than 5% had a significant coronary stenosis and 2% left main stem or triple vessel obstruction [51]. Dyspnoea may also be a marker of physical deconditioning due to low physical activity levels, which is independently associated with CV outcomes [52].

We initially also analyzed on pulmonary embolism as a cause of death. However, because of few events, changes in diagnostic procedures over time and presumed challenges with validity of the diagnosis, we decided to leave this out of the paper.

## Adjustments for confounding factors

It is well known that CV deaths are associated with sex, birth cohort, smoking, occupational exposure to particles, educational level and diagnosis of CV and pulmonary diseases [53,54]. The present study has adjusted for these variables. However, we did not adjust for other known CV risk factors, such as diabetes, obesity, lipids, hypertension, diet, psychological factors, or lung function, which were not available in our study.

The prevalence of breathlessness at a level of effort is higher in women than in men [55], which has been explained by smaller lung volumes in women [56]. The large gender difference in CV mortality has declined during the last 50 years [35], possibly due to interventions of modifiable risk factors in men. Processes related to accelerating ageing are telomere shortening, cell senescence of endothelial cells, diminished cell proliferation and dysfunction of anti-ageing molecules, such as Sirtuin 1. They may play a role in the atherosclerosis and CV deaths [1]. Smoking induces chronic inflammatory responses in susceptible individuals which initiate development and rupture of arteriosclerotic plaques and thus coronary heart disease.

## Strengths and limitations

The large number of participants and a long follow up give a large number of person-years at risk and a high statistical power of the study. The reporting of respiratory symptoms on a self-administered questionnaire removed the biased variation caused by interviewers. The associations were observed in never smokers and in persons without known heart diseases at baseline. The recorders of the cause of deaths were unaware of the baseline variables. To our knowledge, no other population-based study has included analysis of so many respiratory symptoms as predictors of mortality from subgroups of CV diseases.

A majority of the self-reported questionnaires are filled out by the invited persons at home. The persons do not need to be capable of attending the survey station, so frail and ill members of the target population do not need to present themselves. However, the prevalence of respiratory symptoms and the reproducibility of the responses tend to vary by smoking habits and educational level, and the prevalence rates may be influenced by small changes in the phrasing of the questionnaire items [27]. This limits the generalization of the findings in our study.

Remissions of respiratory symptoms are, however, frequent [57,58]. A community study in the Netherlands over 43 years showed that remission of dyspnoea normalizes mortality risk [59], which was confirmed in a 45-year follow up study of men in Sweden [48].

A main cause of uncertainty with regard to cause of death is physicians'reporting of diagnosis on death certificates. Since 2005 the national registry of death has used an automatic classification of medical entities (ACME) which reduce the variation between coder [34]. Previous studies of the population in the City of Bergen have shown substantial agreement between mortality statistic and autopsy findings for fatal stroke and coronary deaths. Cohen's kappa coefficients were 0.78 for stroke and 0.80 for coronary deaths [37].

The 95,705 individuals included in the final analyses represented about 60% of the target population, as we included only individuals with responses to a number of questionnaire items on possible confounding variables. This may be a weakness of the study, as there is a risk that non-participations bias. However, it is hard to delineate how this will impact associations between respiratory symptoms and CV deaths.

A further limitation of our study was the lack of adjustments for potential confounders, as described above. Previous population surveys [48,59] taking body mass index and lung function results into account in Cox proportional hazard regression still show an association between breathlessness and CV death.

Unspecified causes of deaths like other heart diseases and other circulatory diseases were also associated with respiratory symptoms in a dose response matter. The pathogenesis of these groups of diseases can be both inflammatory processes, atherosclerosis with arterial stiffness and aging with telomere shortening, cell senescence of endothelial cells and diminished cell proliferation.

## Implications

Recording of respiratory symptoms is simple, cheap and applicable to most people. We have shown that breathlessness on effort are important predictors of early deaths and mortality for all subgroups of CV diseases.

Persons presenting in a primary care with chronic breathless are frequent and has a wide variety of underlying diagnoses, but large groups are airflow obstruction, CV and circulatory diagnosis, obesity and psychogenic disorders. It is important to highlight the need to establish diagnoses in those persons who seek health care and ensure intervention such as increased physical activity, smoking cessation, psychologic guidance, control of other risk factors and medication. A routine pulmonary and CV check-up should be offered persons with moderate and severe breathlessness which include a thorough physical CV-pulmonary examination, blood pressure, spirometry, chest X-ray, ECG, cholesterol, blood sugar, low-density lipoprotein, B-type natriuretic peptide and if necessary, referral to prolonged ECG and blood pressure. measurements, echocardiography, exercise stress test and coronary angiography. Future studies should examine the most effective approach to diagnose and how to treat people with self-reported breathlessness on effort.

## Conclusion

Breathlessness on effort was a robust indicator of early deaths of all subgroups of CV diseases. The severity of breathlessness increased the risk for such deaths.

## Supporting information

**S1 Table. Descriptive statistics for participants at different stages according to response to questionnaires.**
(PDF)

**S2 Table. Prevalence of baseline respiratory symptoms overall and in never smokers, and according to cardiovascular causes of death.**
(PDF)

**S3 Table. After exclusion of patients with known heart disease.** Hazard ratios with 95% confidence intervals and p-values according to cause of death, multivariable proportional hazards regression analysis adjusted for sex, education, occupational exposure to gas/dust and birth cohort (n = 90,316).
(PDF)

**S4 Table. After exclusion of patients with known heart disease or lung disease.** Hazard ratios with 95% confidence intervals and p-values according to cause of death, multivariable proportional hazards regression analysis adjusted for sex, education, occupational exposure to gas/dust and birth cohort (n = 72,339).
(PDF)

## Author Contributions

**Conceptualization:** Knut Stavem, Amund Gulsvik.

**Data curation:** Knut Stavem, Amund Gulsvik.

**Formal analysis:** Knut Stavem, Henrik Schirmer, Amund Gulsvik.

**Investigation:** Knut Stavem, Amund Gulsvik.

**Methodology:** Knut Stavem, Henrik Schirmer.

**Project administration:** Amund Gulsvik.

**Resources:** Knut Stavem, Amund Gulsvik.

**Software:** Knut Stavem.

**Supervision:** Amund Gulsvik.

**Validation:** Knut Stavem, Henrik Schirmer.

**Visualization:** Knut Stavem, Henrik Schirmer, Amund Gulsvik.

**Writing – original draft:** Knut Stavem, Amund Gulsvik.

**Writing – review & editing:** Knut Stavem, Henrik Schirmer, Amund Gulsvik.

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
