## [Decision Letter · Decision Letter 0]

7 Jul 2022

PONE-D-22-14710Respiratory symptoms and cardiovascular causes of deaths: a population-based study with 45 years of follow-upPLOS ONE

Dear Dr. Knut Stavem!

Thank you for submitting your manuscript to PLOS ONE. After careful consideration, we feel that it has merit but does not fully meet PLOS ONE’s publication criteria as it currently stands. Therefore, we invite you to submit a revised version of the manuscript that addresses the points raised during the review process. I have with great interest read your manuscript. Before we can make a final decision about its future, there are certain revisions that has to be made.

I think the aim is relevant – to determine whether respiratory symptoms increase the mortality (risk) of different cardiovascular diseases. Keep the manuscript to that aim!! Delete all the primary analyses regarding SES, occupational exposures, education, etc., etc. They are also of interest, of course, but in another paper. But of course, keep them as potential confounders, but do not present primary mortality estimates

The table are too many, and difficult to interpret. If you skip the primary analysis of the confounders (and just mention that the analyses are adjusted for occupation, etc, etc), then you can merge some of the tables.

The manuscript has also been read by three experienced referees. Their comments are attached, and I wish that you reply point-by-point to their comments and to my comments above.

We look forward to receiving your revised manuscript.

Kind regards,

Kjell Torén, MD, PhD

Academic Editor

PLOS ONE

Journal Requirements:

Reviewers' comments:

Reviewer's Responses to Questions

**Comments to the Author**

1. Is the manuscript technically sound, and do the data support the conclusions?

Reviewer #1: Yes

Reviewer #2: Yes

Reviewer #3: Partly

2. Has the statistical analysis been performed appropriately and rigorously? 

Reviewer #1: Yes

Reviewer #2: Yes

Reviewer #3: Yes

3. Have the authors made all data underlying the findings in their manuscript fully available?

Reviewer #1: No

Reviewer #2: No

Reviewer #3: Yes

4. Is the manuscript presented in an intelligible fashion and written in standard English?

Reviewer #1: Yes

Reviewer #2: No

Reviewer #3: Yes

5. Review Comments to the Author

Reviewer #1: The aim of the study was to determine the association between respiratory symptoms and death from cardiovascular (CV) during 45 years follow-up in a pooled sample of four Norwegian cohorts including 95,704 participants. Assessing CV deaths and breathlessness is considered relevant to assess the relationship between this common symptom and one of the main causes for death in the general population. The paper is well written and considered to be accessible also for non-specialists. The paper presents interesting findings, and is considered suitable for publication after a minor revision. Please consider the suggestions listed below each section:

Abstract:

Line 3: Consider to replace the word causes with diseases in this sentence to be more clear.

Line 6 : Add “by using”: cardiovascular deaths by using Cox regression analysis……

Line 25 Please consider rephrasing the abstract conclusion to be even more precise. If I am not mistaken the study did not assess if breathlessness is an early marker of CV deaths, but whether breathlessness is associated with CV deaths (as stated in the aim). Alternatively, consider stating state: Breathlessness could be an early marker of CV deaths? (or similar).

Overall: Please omit the space between number and %: 18% instead of 18 %.

Introduction:

The aim is properly placed in the context of the previous literature, which is presented well. The data analyses are considered to support the claims. There does not seem to be a need for detailed protocols and algorithms as supporting information online.

Line 45: Please provide a reference for this statement.

Line 57: should this be CV deaths?

Line 62: consider writing: that were not adjusted for…

Methods:

The details of the methodology are considered to be sufficient to allow the experiments to be reproduced. Nevertheless authors are encouraged to include some brief information in the manuscript regarding the follow-up time, although a reference (19) is provided. Information is included in the follow-up and census data section, but what was the range or the shortest follow-up time?

Results:

The results and tables are well presented and easy to understand for the reader.

Line 229: please omit the bracket after word sex.

Line 234: please consider commenting this finding in the discussion. Could it be due to a healthy worker effect?

Line 245: should this sentence read: myocardial infraction declined with increasing score of breathlessness and wheeze?

Line 258: Please state the direction of this association (higher education seems to be protective).

Line 287: Please insert “was” and “stratified analyses”: lung embolism was excluded in the stratified analyses due to few persons.

Line 296: same as above.

Line 300 to 302: Please consider rephrasing this sentence as it is a bit difficult to understand.

Line 308: Please consider defining early death.

Discussion:

The discussion is balanced and well written, please find some minor suggestions below

Line 361: insert the word “were”: The respiratory

questions were more or less identical to those in the present study.

Line 373: should this read: previous cardiopulmonary disease?

Line 384: sample size of that study?

Line 398: Please consider replacing “are” with “is”: a causal verification is lacking.

Line 403: Replace “have” with “has”.

Line 417: is a reference missing here”?

Line 438-439 The meaning of this sentence: “However, the variability of response…” is not quite clear, please adjust.

An original data deposited in not made available in repositories due to ethical limitations placed by the Regional Committees for Medical and Health Research Ethics in Norway. A minimal data set can be made available for scientific analysis on request, provided that the respective research institution proofs handling of the data strictly in accordance with ethical regulations (written ethics protocol, full compliance with the Declaration of Helsinki).

The study seems to conform well with the STROBE guidelines.

Reviewer #2: Comments to authors.

Thank you for a good manuscript and an exciting study. Research with this type of data is very much needed. The topic is a relevant addition to the existing literature with an impressive amount of data and studied participants giving great strength to the study!

I have some questions about the article:

The authors should consider some language review since the grammar is questionable throughout the manuscript.

Other main areas of concern are considering the tables, which are too large, not self-explanatory and include too much information with limited value (the “other” headings and more). More information is needed concerning the scales used.

1. Abstract:

p2 row 6: insert the word “using”. …using cox analysis

2. Information is needed in the abstract about what scale has been used for assessing breathlessness that has been used (maybe on row 5). (an adaption of the mMRC). As it is now, it is unclear what score three and score two mean. Rows 14-24 are a little vague.

Introduction

1. P4 row 46

2. Should write chronic obstructive pulmonary disease instead of lung disease?

3. Row 50 “No studies are available of studies of” … , strange language.

Methods results and discussion:

1) The section about the study population is hard to follow and understand. Could it be clearer to use some flow diagram or graphic?

2) I need more information concerning the scale used to assess breathlessness. The question asked is vital to interpreting the results. My suggestion is that the table presented in the supplementary material should be included in the manuscript.

3) It is hard to find more thorough information on the “Norwegian respiratory questionnaire” online or elsewhere; the cited sources do not really give any more information. Especially citation 24. However, some information can be found in citation 22. However, this manuscript would benefit from a more comprehensive description of the questionnaires since the symptom questions are the backbone of this study.

4) I lack information on how many were lost to follow-up? ( it is said that 231 emigrated without any available date, but how many were lost all in all? ).

5) More information is needed in the manuscript concerning establishing the cause of death. A smaller section is found in the discussion, but the information is also required in the methods section. (see also points 6-9)

6) What is the validity of the registry of causes of death in this setting? How are they obtained? (through autopsy or clinically only?) Has any previous study examined the agreement between the registered cause of death and the “real” cause? (information needed in the methods section and not only in the discussion)

7) Starting on row 338, the authors discuss pulmonary embolism as a cause of death; it seems like the authors had expected more cases. But what would an expected number of deaths attributed to pulmonary embolism be? How common should death from PE be in an average population? The authors argue that better diagnostics would potentially lead to a higher number, but at the same time argue that the causes of death registry are reliable. Suppose the causes of death are not established by autopsy. In that case, it is plausible that most actual pulmonary embolisms would be categorised as something else (perhaps as a myocardial infarction) by the clinician responsible for the death certificate. This finding then raises concerns about the actual reliability of the causes of the death registry in Norway and then also raises concerns about the outcome variables in this study.

8) Should pulmonary embolism be removed and only mentioned briefly in the text? I do not find that it gives that much important information as it is now – mainly since cases are very few and there are questions concerning its validity.

9) It is unclear if only one cause of death was used in this material? Were secondary or tertiary causes of death available? How was the leading cause of death established? If several simultaneous reasons were present, how was this handled?

In my view, myocardial infarction is often stated as the primary cause of death even when there is no known cause; the actual underlying condition is often more interesting but is often expressed as a secondary or tertiary cause of death. This would jeopardise the reliability of the outcomes. For example, if an individual with severe pulmonary neoplasm in palliative care ultimately died from a cardiovascular condition, wouldn’t the pulmonary neoplasm be the most relevant cause of death? Any thoughts about how this was handled?

10) On the same note, even though this study focuses on cardiovascular disease, it would be interesting to see the proportions of different causes of death in the population. Was cardiovascular disease prevalent in an expected ratio of all deaths?

11) It is unclear to me how adjustments were handled and selected. Were all factors really adjusted for the same variables? As this is a very complex analysis, it is plausible that there is a complex interplay between different factors that could include confounding effects on the analyses. Performing a structured analysis of possible confounders could be of great importance. (such as http://www.dagitty.net/ ). There is a considerable risk that confounders could influence these results, which should be addressed thoroughly. More information on the adjustments is needed within the tables, preferably so that one can see precisely which factors are adjusted for what.

12) Could results be visualised better? Kaplan Mayer or cumulative incidence curves?

13) More work is needed with the tables; they are too large, a lot of information is lacking, and I believe they include too much information. (see my points 13-16 for details)

14) More information is needed within the tables (such as explanations of abbreviations, headings, detailed information on adjustments etc.). The tables are also too large. The tables should stand on their own. As it is now, the reader must always refer to the text to understand the different headings.

15) I am not sure that the headings “other heart”, “other vascular”, and “other ischemic” are needed within the tables. The valuable information from such categories is limited, and I suggest that these are removed with the added benefit of more readable tables. Then there is no need to specify the different headings within the table, which I believe is otherwise needed. The “other” headings are not even referred to in the text. Some explanation on why these are included and what information the authors believe they can get from them is needed in case they are not removed.

16) What are “myocardial infarction – yes/no” and “other heart disease yes/no” as stated in the tables? Is it a previous myocardial infarction/heart disease? What is the value of having these added to the tables?

17) Could educational level be dichotomised in some way to merge the categories?

18) The row “Total” in table 3 is strange. What is meant by this? Is there not a total score for each of the objects within the table – then why choose to include only one? My suggestion is to remove it.

19) Row 389-399: Interesting discussion, but it seems these rows would be better in another section of the manuscript, perhaps below “Main findings” instead, as these are discussed.

Reviewer #3: Dear Editor,

Thank you for the opportunity to review the manuscript: ” Respiratory symptoms and cardiovascular causes of deaths: a population-based study with 45 years of follow-up” by Knut Stavem et al.

The manuscript is well written and discusses an important question on breathlessness symptoms and risk for CVD - death in a large population. These symptoms are very frequent in clinic and these results emphasize the importance of investigation of the causes of breathlessness.

There are some concerns as follows:

In methods: There is a large number of individuals that weren't included in the final analyses. A drop from 158702 individuals to final 95704 was registered. Possible impact of this drop-out should be discussed as a weakness in this study. Moreover, the lack of information on diseases increasing risk for cvd-death as diabetes, and hypertension is also a weakness with this study not mentioned by authors.

The start of the observation: It is not clear to me why the authors didn’t choose the date of filling or left questionnaires for start date of the observation

It is not quite clear what were the procedures in the inclusion for each cohort and whether the questionnaires were filled home or in situ in the study center? Did they have any visit or help on filling in the questionnaires? This should be stated in the methods.

Results: In general, the results are somehow difficult to follow because of the amount of information and tables. The authors should consider concentrating on results important on answering the research question and in specific breathlessness and cvd-mortality. Some of the tables can be part of supplementary material including table 7 and 8.

In Table 7, it is difficult to understand how many events there are in each strata. I suggest a column with information on event for each row.

Discussion: End of the first paragraph” The associations increased with the severity of breathlessness, and the pattern of associations was similar in women and in never smokers”- This is not true as in most of the analyses it wasn’t shown that having 4 symptoms was associated with higher risk. Consider re-writing that.

6. PLOS authors have the option to publish the peer review history of their article (what does this mean?). If published, this will include your full peer review and any attached files.

Reviewer #1: **Yes: **Anne Kristin M. Fell

Senior physician, Associate Professor

Dep. of Occupational and Environmental Health, Telemark Hospital, Skien

Dep. of Community Medicine and Global Health, Institute of Health and Society, University of Oslo

Norway

Reviewer #2: No

Reviewer #3: No

---

## [Editor Report · Decision Letter 1]

10 Oct 2022

Respiratory symptoms and cardiovascular causes of deaths: a population-based study with 45 years of follow-up

PONE-D-22-14710R1

Dear Dr. Stavem,

We’re pleased to inform you that your manuscript has been judged scientifically suitable for publication and will be formally accepted for publication once it meets all outstanding technical requirements.

Kind regards,

Kjell Torén, MD, PhD

Academic Editor

PLOS ONE
---

## [Editor Report · Acceptance letter]

12 Oct 2022

PONE-D-22-14710R1 

Respiratory symptoms and cardiovascular causes of deaths: a population-based study with 45 years of follow-up 

Dear Dr. Stavem:

I'm pleased to inform you that your manuscript has been deemed suitable for publication in PLOS ONE. Congratulations! Your manuscript is now with our production department. 

Kind regards, 

on behalf of

Dr. Kjell Torén 

Academic Editor

PLOS ONE